# The Gut Microbiota Profile in Children with Prader–Willi Syndrome

**DOI:** 10.3390/genes11080904

**Published:** 2020-08-07

**Authors:** Ye Peng, Qiming Tan, Shima Afhami, Edward C. Deehan, Suisha Liang, Marie Gantz, Lucila Triador, Karen L. Madsen, Jens Walter, Hein M. Tun, Andrea M. Haqq

**Affiliations:** 1HKU-Pasteur Research Pole, School of Public Health, University of Hong Kong, Hong Kong 999077, China; pengye@connect.hku.hk (Y.P.); suishal@connect.hku.hk (S.L.); 2Department of Pediatrics, University of Alberta, Edmonton, AB T6G 2R3, Canada; qtan3@ualberta.ca (Q.T.); triador@ualberta.ca (L.T.); 3Department of Agricultural Food & Nutritional Science, University of Alberta, Edmonton, AB T6G 2R3, Canada; afhami@ualberta.ca (S.A.); deehan@ualberta.ca (E.C.D.); jwalter1@ualberta.ca (J.W.); 4Biostatistics & Epidemiology Division, RTI International, Durham, DC 27709, USA; mgantz@rti.org; 5Department of Medicine, University of Alberta, Edmonton, AB T6G 2R3, Canada; kmadsen@ualberta.ca; 6Department of Biological Sciences, University of Alberta, Edmonton, AB T6G 2R3, Canada; 7APC Microbiome Ireland, School of Microbiology, and Department of Medicine, University College Cork–National University of Ireland, T12 YN60 Cork, Ireland

**Keywords:** Prader–Willi syndrome, gut microbiota, bacteria, fungi, diet, hyperphagia, obesity, cross-sectional

## Abstract

Although gut microbiota has been suggested to play a role in disease phenotypes of Prader–Willi syndrome (PWS), little is known about its composition in affected children and how it relates to hyperphagia. This cross-sectional study aimed to characterize the gut bacterial and fungal communities of children with PWS, and to determine associations with hyperphagia. Fecal samples were collected from 25 children with PWS and 25 age-, sex-, and body mass index-matched controls. Dietary intake data, hyperphagia scores, and relevant clinical information were also obtained. Fecal bacterial and fungal communities were characterized by 16S rRNA and ITS2 sequencing, respectively. Overall bacterial α-diversity and compositions of PWS were not different from those of the controls, but 13 bacterial genera were identified to be differentially abundant. Interestingly, the fungal community, as well as specific genera, were different between PWS and controls. The majority of the variation in the gut microbiota was not attributed to differences in dietary intake or the impact of genotype. Hyperphagia scores were associated with fungal α-diversity and relative abundance of several taxa, such as *Staphylococcus*, *Clostridium*, *SMB53*, and *Candida*. Further longitudinal studies correlating changes in the microbiome with the degree of hyperphagia and studies integrating multi-omics data are warranted.

## 1. Introduction

Prader–Willi syndrome (PWS) is a complex genetic disorder known as the most common syndromic cause of childhood obesity. It is caused by the lack of expression of paternal alleles in 15q11-13, with an incidence of 1 in 10,000 to 15,000 live births worldwide [1,2]. Three common forms of PWS are deletion (65–75%), maternal uniparental disomy (UPD; 20–30%), and imprinting defects (ID; 1–3%) [3]. Affected children typically experience excessive and rapid weight gain between 12 months and 6 years of age, which coincides with an onset of hyperphagia (abnormally increased appetite). Hyperphagia is a critical diagnostic feature of PWS, characterized by a deficit in satiety, food preoccupations, and problematic food-seeking behaviors [4]. Patients with PWS are at elevated risk of developing morbid obesity and associated life-threatening complications, unless eating is externally controlled. Although symptom severity varies, hyperphagia generally requires individuals to have a restricted lifestyle with constant supervision, which hinders their independence and significantly affects the quality of life of these patients, as well as their families [5].

Recent studies suggest that PWS-associated obesity is accompanied by alterations in the gut microbiota, offering new insights into the pathophysiology of obesity in patients with PWS. Zhang et al. were the first to demonstrate that children with PWS-associated obesity (*n* = 17) and those with idiopathic obesity (*n* = 21) shared similar dysbiotic gut microbiota features, characterized by a higher diversity and abundance of toxin-producing and potentially pathogenic bacteria, compared to lean controls [6]. In addition, the colonization of germ-free mice with a microbial community from a subject with PWS transferred some aspects of the obesity phenotype, such as higher inflammation and larger adipocytes. The authors concluded that the gut microbiota may play an important role in inducing and promoting obesity and metabolic complications in PWS, and hence could be a potential therapeutic target. Moreover, work from Olsson et al. recently showed that, compared to obese controls, the fecal microbiota of adults with PWS-associated obesity was characterized by higher phylogenetic diversity, and was associated with markers of insulin sensitivity [7]. In addition, the gut microbial communities in patients with PWS were more similar to their non-PWS parents than those of obese controls [7].

Characterization of the gut microbiota in individuals with PWS is an important initial step in understanding the role of the microbiome in the course and outcome of hyperphagia, obesity, and metabolic deterioration in PWS. The only two observational studies characterizing the gut microbiome in PWS have reported inconsistent findings, which may be attributable to the differences in age and ethnicity of their subjects and in the specific control groups included. Additionally, no studies have yet assessed the gut microbiota profile of PWS in childhood populations in North America. Although fungi have been shown to be implicated in health and diseases, including metabolic disorders [8], differences in the fungal community between individuals with and without PWS have not been assessed at all. Furthermore, little is known about whether gut microbiota in overweight or obese (OWOB) subjects are distinct from those in normal weight (NW) subjects within the PWS population. To gain insight into the potential links between the microbiome and clinical manifestations of PWS, additional investigations are needed to define the microbial signatures specific to PWS, as well as to the OWOB within PWS. In this study, we profiled the fecal microbiota in children with PWS and compared it to that of healthy matched controls, while also assessing for associations between microbial composition and hyperphagic symptoms. As hyperphagia in PWS likely results in nutrient intake differences between PWS and controls, we further investigated the relationship between diet and fecal microbial composition.

## 2. Materials and Methods

### 2.1. Subjects

This cross-sectional study was completed in Edmonton, Alberta, Canada, from February 2017 to July 2018, and was approved by the University of Alberta’s Research Ethics Boards (Pro00069925). A prior written/verbal informed consent was obtained from the children and their parents or legal guardians to participate in the study. Children with genetically confirmed PWS (*n* = 25; aged 3–17 years) were recruited from the Pediatric Endocrine Clinic at the Stollery Children’s Hospital, as well as remotely through collaboration with the Foundation for Prader–Willi Research (Canada/United States) and the PWS Association (United States). Controls (CON) (*n* = 25) included age-, sex-, and body mass index (BMI) *z*-score-matched participants, who were recruited by advertising on bulletin boards and via e-mail distribution lists at the University of Alberta. We collected a fecal sample and data on hyperphagia, dietary intake, and anthropometric measurements. Additional parent-reported information including gestational age, delivery mode, and infant feeding methods; use of probiotics and proton pump inhibitors, as well as medication was also collected. Exclusion criteria were (a) a pre-existing condition that could affect body weight; (b) use of antibiotics 30 days prior to study, or (c) administration of pre/probiotic supplements or antibiotics. Growth hormone therapy was permitted, as this treatment is commonly used in patients with PWS to counteract their endogenous deficiency of growth hormone.

### 2.2. Assessments

Hyperphagia was assessed using the Hyperphagia Questionnaire, a multi-item, provider-reported instrument widely used in clinical practice that has been specifically designed and validated to capture hyperphagic symptoms in PWS [9]. Each item was rated on a five-point scale (1 = not a problem to 5 = severe and/or frequent problem). The scores were calculated by summing the following items: behavior (items 2, 4, 5, 8, 10); drive (items 1, 3, 6, 9); severity (items 7, 11); and total (items 1 to 11) scores (see example questionnaire in Appendix B).

Dietary intake was assessed using a three-day dietary record; children and parents were asked to provide detailed information on the food and beverages consumed by the children over two non-consecutive weekdays and one weekend day. Analysis of dietary data was performed using Food Processor SQL (Version 11.4, ESHA Research Inc., Salem, OR, USA). Daily intake of energy and macronutrients (carbohydrates, protein, lipids, sugar, dietary fiber, sat/unsaturated fat, and cholesterol) was estimated. For correlation analyses, nutrient intakes were adjusted for total daily energy intake using the nutrient residual method [10].

Weight, height, and waist circumference (WC) were measured following standardized procedures described by the National Health and Nutrition Examination Survey Anthropometry Procedures Manual [11]. Weight was measured to the nearest 0.1 kg using an electronic weighing scale. Height was measured to the nearest 0.1 cm using a wall-mounted stadiometer. WC was recorded to the nearest 0.1 cm with a non-stretch measuring tape between the bottom of the lower rib and the iliac crest. BMI-for-age percentile (BMI %ile) and BMI *z*-scores were calculated and classified according to the World Health Organization BMI-for-age growth standards (for 5–19 years) [12]. Normal weight was defined as a BMI *z*-score ≤ 1 SD, while overweight was defined as >1 to ≤2 SD, and obesity as > 2 SD.

### 2.3. Fecal Sample Collection, DNA Sequencing, and Microbiome Analysis

Subjects were provided with OMNIgene∙GUT kits (DNA Genotek, Inc., Ottawa, CA, USA) and instructed to collect feces at home, and send the sample through expedited mail delivery service on the day of collection. In OMNIgene∙GUT kits, samples were diluted in a proprietary solution, which has been previously shown to keep microbial DNA stable [13]. Within 24 h of receipt, fecal slurries were aliquoted and immediately frozen at −80 °C until being sent to Microbiome Insights Inc. (Vancouver, BC, Canada) for sequencing. DNA was extracted from the fecal homogenates using the PowerMag Soil DNA Isolation Kit (MoBio Laboratories, Carlsbad, CA, USA) per manufacturer’s protocol. PCR amplification of the bacterial 16S rRNA gene targeting the V3–V4 region and the fungal ITS2 gene was performed with dual-barcoded primers, as previously described [14]. Amplicons were sequenced with an Illumina MiSeq using the 300 bp paired-end kit (v.3). Sequences were deposited at the European Nucleotide Archive under PRJEB34398 (16S data) and PRJEB34790 (ITS2 data). Sequences were denoised and taxonomically classified, using Greengenes (v. 13_8) as the reference database. Sequences were clustered into operational taxonomic units (OTUs) using Mothur v. 1.39.5 [15], following the recommended procedure that are illustrated on the mothur website (https://mothur.org/wiki/miseq_sop/) [15].

The potential for contamination was addressed by co-sequencing DNA amplified from specimens, and from four each of template-free controls and extraction kit reagents, processed the same way as the specimens. Two positive controls consisting of cloned SUP05 DNA were also included (number of copies = 2 × 10^6^). Operational taxonomic units were considered putative contaminants (and were removed) if their mean abundance in controls reached or exceeded 25% of their mean abundance in specimens. In addition, OTUs that had fewer than three occurrences (counts) in 10% of the samples were considered noise, and were removed prior to statistical analysis.

### 2.4. Statistical Analyses

The significance of differences in dietary intake and diversity between PWS and CON, as well as between the subgroups (i.e., OWOB PWS vs. NW PWS, OWOB CON vs. NW CON, OWOB PWS vs. OWOB CON, and NW PWS vs. NW CON) were tested using a two-sided Wilcoxon test, using the *wilcox.test* function in R package stats v3.5.1 and Dunn’s tests in R package *dunn.test* v1.3.5 with Bonferroni correction, respectively. To assess α-diversity, the Shannon, Simpson, and Chao1 indices were calculated for the overall bacterial community, as well as for major bacterial and fungal phylum using the diversity function in R package vegan v2.5-6 and the chao1 function in R package fossil v0.3.7, respectively. Inter-subject β-diversity, represented by Bray–Curtis, distance-based dissimilarity, was calculated for both the bacterial and the fungal communities using the *vegdist* function in R package vegan v2.5-6. Permutational multivariate analysis of variance (PERMANOVA) was used to assess differences in community structure, with group as the main fixed factor and with 9999 permutations for significance testing, using the *adonis* function in R package vegan. Canonical correspondence analysis (CCA) using the *cca* function with the *envfit* function in R package vegan (999 permutations) was used to assess the association between subject characteristics and dissimilarity between the microbial communities mentioned above, using OTU-level data.

Linear discriminant analysis effect size (LEfSe; https://huttenhower.sph.harvard.edu/galaxy/) was determined, in order to identify differentially abundant bacterial and fungal genera between PWS and CON, as well as between the two subgroups within PWS and CON (i.e., OWOB PWS vs. NW PWS, and OWOB CON vs. NW CON); *p*-values < 0.05 were considered statistically significant for the first step (Wilcoxon tests), while Linear Discriminant Analysis (LDA) minimal threshold was set at 2. as previously described [16]. Additionally, the random forest algorithm provided in *RandomForestClassifier* of the Python module *sklearn.ensemble* was used to construct a classification model using genus-level relative abundance data. The original model was built by including all bacterial or fungal genera, with variance in the stratified samples. Following that, genera with a Gini importance greater than or equal to the 90% quantile in each original model were selected to construct the final model, of which performance of classification was assessed using 5-fold cross-validation.

For genera with occurrence frequency greater than 50% among all subjects, Spearman’s correlations were calculated using the *cor.test* function in R package stats v3.5.1, in order to assess associations between diversity indices and taxa abundance with dietary intakes in the whole dataset, and hyperphagia scores in both the whole dataset and within the PWS group. The *p*-values were corrected for multiple testing using the Bonferroni method, and resulting *q*-values of less than 0.1 were regarded as statistically significant. Binary logistic regression models in SPSS were used to investigate the associations between adjusted dietary carbohydrate intake (adj-CHO) and hyperphagia scores, and abundances of the marker taxa were detected by random forest classifiers while adjusting for the group and its interaction. In the models, adj-CHO values, hyperphagia scores, and the microbial abundances were dichotomized into “above-the-median” and “below-the-median”.

In addition, Spearman’s correlations calculated using the *cor.test* function in R package *stats* v3.5.1 were also used to construct bacterial–fungal inter-kingdom networks for microbial families with occurrence frequency greater than 50% in each subgroup. All taxon pairs with Spearman’s Rho ≥ 0.3 or ≤ −0.3 with *p*-values < 0.05 were visualized using Cytoscape [17].

## 3. Results

Next-generation sequencing of PCR-generated amplicons of the V3–V4 region of bacterial 16S rRNA gene and fungal ITS2 gene resulted in an average of 57,133 and 5732 quality-filtered reads per sample, respectively. Samples for the ITS2 gene containing fewer than 1000 sequences were excluded from downstream analyses (six from PWS, seven from CON). After quality control, a total of 2498 bacterial OTUs and 255 fungal OTUs were identified in the samples. These OTUs were annotated and assigned taxonomy ranging from phylum to species.

### 3.1. Gut Bacterial Communities in PWS

No significant difference was observed between PWS and CON, or between the subgroups for overall bacterial α-diversity (Appendix A). However, more targeted analyses within bacterial phyla showed higher Chao1 richness of *Actinobacteria* in PWS compared to CON, as well as in NW PWS when compared to NW CON. In addition, fecal samples from subjects with PWS displayed a lower α-diversity of *Proteobacteria* than healthy controls assessed by the Shannon and Simpson indices, which both estimate species richness and evenness with the Shannon index giving more weight on evenness and the Simpson index on richness, respectively [18]. Between OWOB subgroups, the Simpson index of *Proteobacteria* was lower in PWS OWOB than in the matched CON (Appendix A).

PERMANOVA showed no significant difference in the overall bacterial community structure between PWS and CON subjects (*p* = 0.389, Figure 1A). CCA results suggest potential association between weight (*p* = 0.016), sex (*p* = 0.001), age (*p* = 0.004), and calorie-adjusted dietary factors (fiber (*p* = 0.008), protein (*p* = 0.017), cholesterol (*p* = 0.001), and sugar (*p* = 0.040)) and bacterial community structure (Figure 1C). There were no significant differences of the overall bacterial community structure between the genetic subtypes of PWS (deletion versus UPD form (*p* > 0.05)).

### 3.2. Gut Fungal Communities in PWS

The α-diversity of fungal communities between PWS and CON and between the subgroups was not different. However, within the phylum *Basidiomycota*, the Shannon index suggests higher species diversity for children in the OWOB PWS group than those in the OWOB CON group (Appendix A). Interestingly, the fungal communities of PWS subjects were significantly different from those of the match CON (*p* = 0.009, PERMANOVA, Figure 1B). PWS (*p* = 0.001), age (*p* = 0.002), and hyperphagic drive score (*p* = 0.012) might be factors associated with the observed differences (CCA, Figure 1D), whereas we did not find any association between body weight and fungal community (*p* > 0.05). No significant differences of the overall fungal community structure were found between the genetic subtypes of PWS (deletion versus UPD form (*p* > 0.05)).

### 3.3. Abundance of Certain Taxa Is Associated with PWS

LEfSe identified differentially abundant bacterial taxa between PWS and CON, and also between the subgroups in PWS. The genus *Prevotella* showed a higher abundance in children with PWS, whereas *Oscillospira* and unclassified *Enterobacteriaceae* genera were more abundant in healthy controls (Figure 2A). Within PWS, the genera *Sutterella* and *Clostridium* were more abundant in OWOB, while higher abundance of *Turicibacter* and unclassified *Barnesiellaceae* were observed in NW subjects (Figure 2C). Among the fungal taxa, the genera *Candida*, *Mrakia*, and unclassified *Agaricomycetes* and *Basidiomycota* were in higher abundance in PWS compared with CON; whereas CON showed higher fecal abundances of *Saccharomyces* (Figure 2B). Within PWS, unclassified *Basidiomycota* was more abundant in OWOB compared to NW (Figure 2D). 

We also constructed random forest classifiers to further identify taxa discriminating PWS from CON (Appendix A). All final models showed high discriminatory power, as indicated by an area under the receiver operating characteristic curve of ≥0.80 (Appendix A). The algorithm detected 13 bacterial genera, with only four of them belonging to the same genera, identified by the differential analyses above. The additional OTUs identified by this method were related to *Staphylococcus*, *Propionibacterium*, *Akkermansia*, *Haemophilus*, unclassified *RF39* and *Christensenellaceae*, *Anaerostipes*, *Dorea*, and *SMB53* (Figure 2A and Figure 3A). Conversely, most of the selected fungal taxa (*Mrakia*, *Candida*, *Saccharomyces*, and unclassified *Agaricomycetes* and *Basidiomycota*) in the model were previously identified by LEfSe, except for the unclassified *Ascomycota* (Figure 2B and Figure 3B). The differential abundances of both bacterial and fungal genera under the different random forest models tested are summarized in Appendix A.

### 3.4. Gut Microbiota and Dietary Carbohydrate Intake in the Whole Dataset

Sugar intake was found to be positively associated with the relative abundance of an unclassified *Christensenellaceae* (Spearman’s Rho (ρ) = 0.44, *q*-value = 0.071) and bacterial α-diversity (Shannon index: ρ = 0.38, *q*-value = 0.020; and Simpson index: ρ = 0.37, *q*-value = 0.027; Appendix A). Dietary fiber intake was negatively correlated with the relative abundance of *Saccharomyces* (ρ = 0.40, *q*-value = 0.055) and fungal Shannon diversity (ρ = −0.44, *q*-value = 0.023; Appendix A). In addition, carbohydrate intake (CHO) was positively associated with the relative abundance of *Saccharomyces* (ρ = 0.45, *q*-value = 0.021; Appendix A), while also positively associated with bacterial Shannon diversity (ρ = 0.32, *q*-value = 0.066; Appendix A).

Dietary analysis showed no significant differences in dietary protein and fat intakes between PWS and CON (*p* > 0.05), whereas CHO was significantly lower in PWS compared to CON (*p* = 0.002; Table 1). We used logistic regression models to investigate whether differential abundances of the taxa between PWS and CON detected by the random forest models were associated with differences in adjusted dietary CHO (adj-CHO), while adjusting for the group and their interaction (adj-CHO * group). We found that a lower abundance of unclassified *Christensenellaceae* was significantly associated with lower adj-CHO intake (adjusted odds ratio = 0.14 (95% CI: 0.02–0.94); *p* = 0.043), but was not associated with the group (adjusted odds ratio = 0.61 (95% CI, 0.09–4.02); *p* = 0.608). In addition, there was no significant association between the interaction of adj-CHO or PWS and any taxon (Appendix A).

### 3.5. Gut Microbiota and Hyperphagia in PWS

Since PWS is characterized by hyperphagia, we also explored associations using Spearman’s correlations between hyperphagic symptoms and fecal microbiota composition. In the whole study dataset, no significant correlations were observed between hyperphagia and bacterial features (*q* > 0.1). However, severity scores were negatively correlated with unclassified *Ascomycota* (ρ = −0.47, *q*-value = 0.015), while behavior scores were correlated with an unclassified fungal taxon (ρ = −0.45, *q*-value = 0.023; Figure 4A). Furthermore, the behavior score was positively associated with fungal Simpson diversity (ρ = 0.42, *q*-value = 0.029; Figure 4B). When further investigating the correlations within the PWS group, the relative abundance of the unclassified fungi was observed again to be negatively correlated with the behavior score (ρ = −0.65, *q*-value = 0.014, Figure 4C). Strong associations were also found between behavior score and fungal α-diversity (Shannon diversity index: ρ = 0.71, *q*-value = 0.002; and Simpson diversity index: ρ = 0.76, *q*-value < 0.001; Figure 4D). In addition, the total score was positively associated with fungal α-diversity (Shannon diversity index: ρ = 0.50, *q*-value = 0.089; Simpson diversity index: ρ = 0.57, *q*-value = 0.032; Figure 4D). Using logistic regression models, we found no significant association between hyperphagia scores and microbial abundance (above vs. below median) in the whole dataset (Appendix A).

### 3.6. Inter-Kingdom Ecological Networks in PWS

In addition to different microbial compositions, we observed a difference within the inter-kingdom ecological networks between the PWS and CON groups (Figure 5). Specifically, at the family level, microbial communities in PWS had overall denser bacteria-fungal networks than those of the CON group, as illustrated by an increase in the number of nodes (taxa) and edges (significant interactions) in PWS. Within subgroups, microbial communities in the OWOB group appeared to have fewer bacteria-fungi network than those in the NW group, with a smaller number of nodes (NW CON: 27, OWOB: 23; NW PWS: 38, OWOB PWS: 34) and neighbors (NW CON 3.19, OWOB: 1.83; NW PWS: 3.42, OWOB PWS: 3.18). In addition, there were more positive inter-kingdom interactions than negative interactions within the four subgroups (116 positive interactions vs. 67 negative interactions in total). The ratio of positive to negative interactions was higher in the PWS subjects (2.18 in OWOB PWS, 1.71 in NW PWS) than in the CON (1.33 OWOB CON, 1.53 NW CON), suggesting that the microbial community is more cooperative in PWS than CON. Taken together, these results suggest a complex relationship between the bacteria and fungi in the gut microbiota, and that disease-specific differences are present in PWS.

## 4. Discussion

The present study demonstrates that the fecal microbiota differs between children with and without PWS. In addition to differences in bacterial composition, for the first time, we show that the fecal mycobiota in PWS is also different from controls. The simultaneous analysis of both the bacterial and fungal microbiota enabled the elucidation of differences in the inter-kingdom interactions between patients with PWS and matched CON. By applying abundance analyses, we also identify taxa that are differentially abundant in PWS and CON subjects.

Loss of microbial diversity is a constant finding of intestinal dysbiosis, and seems to be associated with most chronic diseases, including obesity [19,20]. In our study, analysis of overall bacterial and fungal diversity found no difference between PWS and CON. In addition, no difference was found in the bacterial community structure between PWS and CON; however, patients with PWS showed a fungal community structure distinct from the matched controls. These differences might be merely the consequence of the disease, environmental factors, or covariates unrelated to PWS that are not examined here. Garcia-Ribera et al. also recently analyzed the fecal microbiota composition in children with PWS and observed higher phylogenetic diversity in normal weight subjects compared to those who were overweight or obese [21]. Our subgroup analyses, however, found no differences in microbial diversity between OWOB PWS and NW PWS. In contrast to the study by Olsson et al., which reported higher overall phylogenetic diversity in PWS adults with obesity compared to individuals with idiopathic obesity [7], the present study did not detect any difference between the OWOB PWS and OWOB CON groups. The discrepancies between these studies might be caused by differences in participant characteristics (age, diet, geography, socioeconomic status, and ethnicity), sample sizes, and analytic tools that were used. Olsson et al. additionally demonstrated that the phylogenetic diversity in subjects with PWS was similar to that of their non-obese, non-PWS parents, suggesting that the community is not likely shaped by PWS status or obesity in individuals with PWS, but instead by environmental factors. Microbiota diversity is potentially important in gut health and temporal stability; further studies are needed to investigate the compositional, and more importantly, functional diversity in the gut microbiota of individuals with PWS.

The LEfSe and random forest analyses demonstrated that gut bacterial and fungal microbiota were characterized by differences in their compositions in PWS, with certain taxa showing higher or lower abundance in children with PWS compared to controls. Both methods detected a higher abundance of *Prevotella* in PWS compared to CON, while subgroup analyses found its abundance was lower in OWOB PWS relative to NW PWS. There is human evidence linking gut bacteria and appetite, showing that microbiota activity is associated with changes in plasma levels of appetite-regulating hormones, such as glucagon-like peptide 1, peptide YY, and ghrelin [22]. Previously, the abundance of *Prevotella* species was found to be positively correlated with ghrelin, while negatively correlated with leptin in rat models [23]. *Prevotella* are well-known fiber degraders [24], and mechanistic studies suggest microbial metabolites produced during dietary fiber fermentation (i.e., short-chain fatty acids) as an important modulator of host metabolism [22]. Therefore, the increased abundance of *Prevotella* and its correlations with appetite and bodyweight in patients with PWS may deserve further study. However, *Oscillospira*, a genus that has been associated with lower BMI [25,26], was lower in abundance in PWS compared with CON. The RF model identified 10 additional genera, out of which four taxa (*Dorea*, *Akkermansia*, and genera classified in the *RF39* and *Christensenellaceae* families) had been previously reported by Olsson et al. Our study observed a lower abundance of *Dorea* in patients with PWS, which is in agreement with Olsson et al. showing significantly lower *Dorea* in subjects with PWS. Contrary to the enrichment of *Akkermansia* in PWS described in their report, we found a significantly lower abundance of this genus in children with PWS compared to controls. *Akkermansia*, in particular *Akkermansia muciniphila*, has been associated with intestinal integrity, and its abundance has been inversely correlated to obesity and metabolic dysfunction [27,28,29,30,31,32]. The change in the abundance of *Akkermansia* and its correlation with body weight and metabolic health in individuals with PWS might be worth exploring.

For differentially abundant fungi between PWS and CON, LEfSe analysis shows high concordance with results from RF: the same genera (*Mrakia*, *Candida*, *Saccharomyces*, and genera classified in the *Agaricomycetes* and *Basidiomycota* families) were obtained from both methods, with the exception of an unclassified *Ascomycota*. The major pathogenic *Candida* species in humans is *Candida albicans*, which, under normal conditions, behaves like commensal members of the gut microbiota, and its colonization is limited by other bacterial species [33]. Higher abundance of *Candida* might indicate disruptions in the microbial equilibrium, the intestinal mucosal barrier, or host’s innate immune system in children with PWS [34]. Interestingly, a clear reduction in *Saccharomyces* was observed in the overall PWS group, but that could be due to lower consumption of fermented foods, such as bread. Overall, it should be considered that fungi are an overlooked kingdom of the microbiome, and we are only beginning to understand their structure, role, and functions; thus, the biological and clinical significance of observed changes in the gut mycobiome in children with PWS remains to be determined.

*Dialister* and *Bifidobacterium* were consistently enriched in OWOB subjects within the PWS and CON groups, shown in both the random forest models and LEfSe. Higher abundance of *Dialister* has been associated with failure to lose body weight in an intervention program [35]. *Bifidobacterium* is generally reported to be correlated with healthy status, but its effect on body weight is strain-specific, as reported previously in a mouse study [36]. Increased abundance of this genus in OWOB subjects may, therefore, reflect the enrichment of some strains that are able to induce increases in body weight.

Fungi and bacteria coexist in the gut; thus, their interaction networks may be altered in the disease state [37]. Alterations in bacteria–fungi interaction networks have been previously reported in colorectal cancer [38], inflammatory bowel disease [39,40], and ankylosing spondylitis [41]. In this study, we also noted a PWS-specific pattern for the inter-kingdom network. Children with PWS had more densely connected inter-kingdom co-occurrence networks compared to the CON, and those networks were largely positive, suggesting more collaboration between bacterial and fungal communities in PWS. Overall, these findings indicate that inter-kingdom interactions in the gut are an underappreciated, but potentially important component of human disease that deserves more research attention.

Analysis of macronutrient intakes revealed a significantly lower dietary carbohydrate intake in PWS, which might be linked to variation identified above. However, a logistic regression model showed that only unclassified *Christensenellaceae* was significantly associated with CHO. For most of the differentially abundant genera, these factors had no influence, suggesting that diet or PWS per se, or their interaction, was not a strong determinant of microbiota composition, which is in accordance with previous studies [42,43]. One study reported that in healthy Israeli individuals, non-genetic factors (age, sex, BMI, smoking status, and dietary patterns) and genetic factors explain only about 10% of the observed variation in the gut microbiome [42], and another study in northern Germany found that effects of specific environmental factors, such as diet, medication, and anthropometric measurements account for approximately 20% of the variation [43]. Overall, these results suggest that host and environment make only a small contribution to interindividual microbiota variation, implying some unknown sources of variance.

In this study, we report for the first time correlations between hyperphagic symptoms and microbiome features observed in patients with PWS. Specifically, mycobiota diversity was strongly and positively correlated with behavior scores. One possible explanation for this positive association is that these children might consume a greater variety and larger amounts of foods, as a result of bargaining, seeking, stealing, and foraging for food [4]. Further analysis of hyperphagia and fungal genus-level data revealed a negative correlation between the abundance of an unclassified *Ascomycota* and hyperphagia severity scores among the study dataset. The abundance of this fungal taxon was reduced in the PWS group and in OWOB subjects compared to NW subjects within the PWS and the CON groups. These results suggest a link between PWS-specific gut microbial features and hyperphagic symptoms. Further work studying changes in the gut microbiota throughout the lifespan and course of the disease are needed to determine whether diversity and abundance differences exacerbate hyperphagic symptoms, merely as a consequence of disease progression, or potentially driven by environmental factors.

One of the strengths of our study lies in the inclusion of normal weight and overweight/obese patients with PWS. Many children with PWS nowadays are able to maintain a normal body weight, since early diagnosis followed by rapid start of treatment, such as growth hormone replacement therapy and energy-restricted diet, may reduce the risk of early-onset obesity. An added advantage of our study is the analysis of the correlation between hyperphagia and dietary intake data and microbial abundances. Hyperphagia is the most striking clinical feature of PWS, but its symptoms vary greatly among patients. This study is the first to identify microbial taxa significantly associated with hyperphagia in PWS, which may represent a signature of varied appetite regulation. A limitation of this study was the small size of the PWS group, which is a common barrier in all rare disease studies; however, it was large enough for us to study the differences in the gut microbiota in children with and without PWS. We also note that the cross-sectional design limited interpretation of the results, particularly with respect to hyperphagia. Another limitation was that gut microbiota of family members of PWS were not analyzed in this study like in previous studies [7]. Including these data may facilitate understanding of the extent to which genetics contribute to the PWS-associated gut microbiota. In addition, the Hyperphagia Questionnaire, specifically designed for PWS, is not a valid instrument for use in healthy individuals, which might limit the conclusions reached. The species-level profiles generated by the amplicon sequencing methods used in this study were not reliable. Considering the zero-inflated nature of species/OTU-level abundance and the importance of precise taxonomic assignments, we used genus-level or above data in the analyses regarding comparisons of microbial abundances or the correlation between microbial abundances and phenotypes. Using additional methodologies, such as shotgun metagenomic sequencing and high-throughput culturomics, will facilitate strain-level dissection of the microbiome, and will allow us to identify specific functional bacterial and fungal strains and gain mechanistic insight into their role in the onset and progression of hyperphagia in PWS.

Although our study detected differences in the gut microbiota in children suffering from PWS as compared to healthy controls, our analysis does not provide any information as to whether such differences are causal or contributory to disease, rather than a consequence, or if they constitute compensatory beneficial responses. In this respect, it is important to point out that previous research on the role of the gut microbiota in PWS resulted in contradictory findings. Using experiments where human microbiota was transplanted into germ-free mice, Zhang et al. reported a causal contribution of gut dysbiosis to obesity in children with PWS [6], while Olsson et al. suggested that PWS-associated microbial features might play a role in the prevention of metabolic complications [7]. These discrepancies might stem from the inability of human microbiota-associated mice to make accurate predictions about causal claims [44,45]. With the limited evidence, it is too early to attribute any phenotypic effects to the microbiome without more robust investigation. More complex analysis, such as multi-omics and time-series measurements, and an interventionist framework are likely required if we want to approach an assertion about causality [44]. For example, the differences detected by us and the previous studies in microbiota composition could be targeted by nutritional or microbial-based approaches, in an attempt to improve outcomes in PWS patients. Should this be possible, microbiota features are likely to contribute or prevent pathology in PWS.

In conclusion, gut microbiota composition of patients with PWS differs from matched controls. In addition to differences in the relative abundance of certain bacterial taxa, our findings reveal differences in the gut fungal community and bacteria–fungi interkingdom interactions in PWS. Moreover, we identified several interesting links between gut microbes and hyperphagia in PWS. However, further work that includes a time-series of samples over the course of the disease and multi-omics data is required for the evaluation of the role of gut microbiome in the modulation of hyperphagic symptoms in PWS. While the implications of the present findings remain to be refined, they pave the way for the elucidation of characteristics of a PWS-specific gut microbiome, and lay the groundwork needed for future study of microbiota-based treatment strategies for individuals with PWS.

## Figures and Tables

**Figure 1 genes-11-00904-f001:**
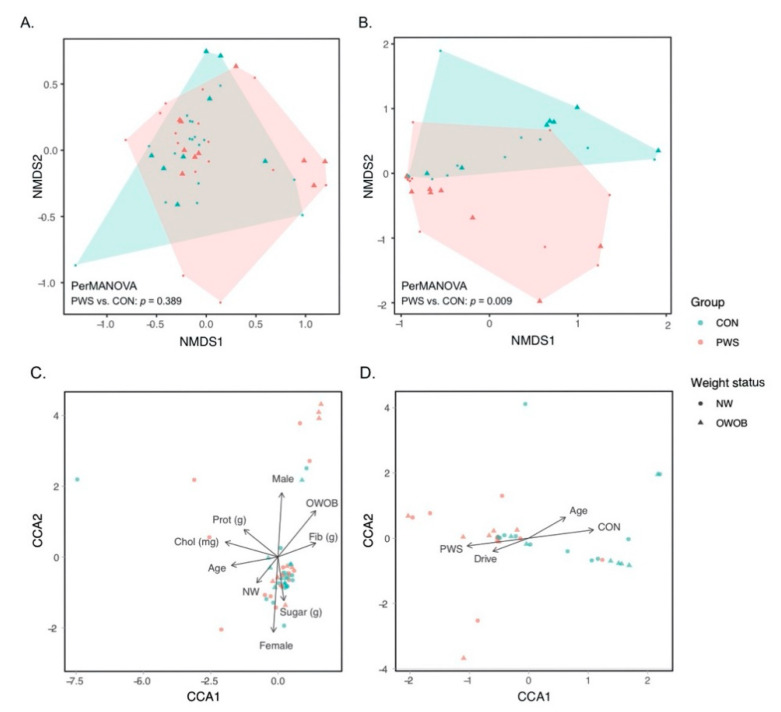
Phylum-level relative abundance in bacterial communities (**A**) and fungal communities (**B**). Linkages between the microbial community and environmental variables. The first two axes of the canonical correspondence analysis (CCA) display bacterial (**C**) and fungal (**D**) community (dots) and phenotypes (arrows) with significant linkage (*p* < 0.05). PWS: Prader–Willi syndrome; CON: control. Arrows show the level and direction of impact of significant factors.

**Figure 2 genes-11-00904-f002:**
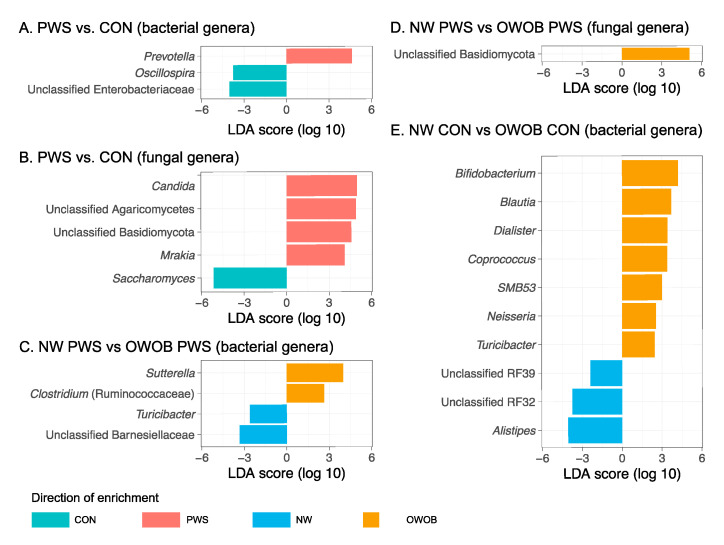
Discriminative features in the linear discriminant analysis effect size (LEfSe) analysis. (**A**) Discriminative bacterial genera in PWS vs. CON. (**B**) Discriminative fungal genera in PWS vs CON. (**C**) Discriminative bacterial genera in NW PWS vs. OWOB PWS groups. (**D**) Discriminative fungal genera in NW PWS vs. OWOB PWS groups. (**E**) Discriminative bacterial genera in NW CON vs. OWOB CON groups. PWS: Prader–Willi syndrome; CON: control. NW: normal weight; OWOB: overweight or obese.

**Figure 3 genes-11-00904-f003:**
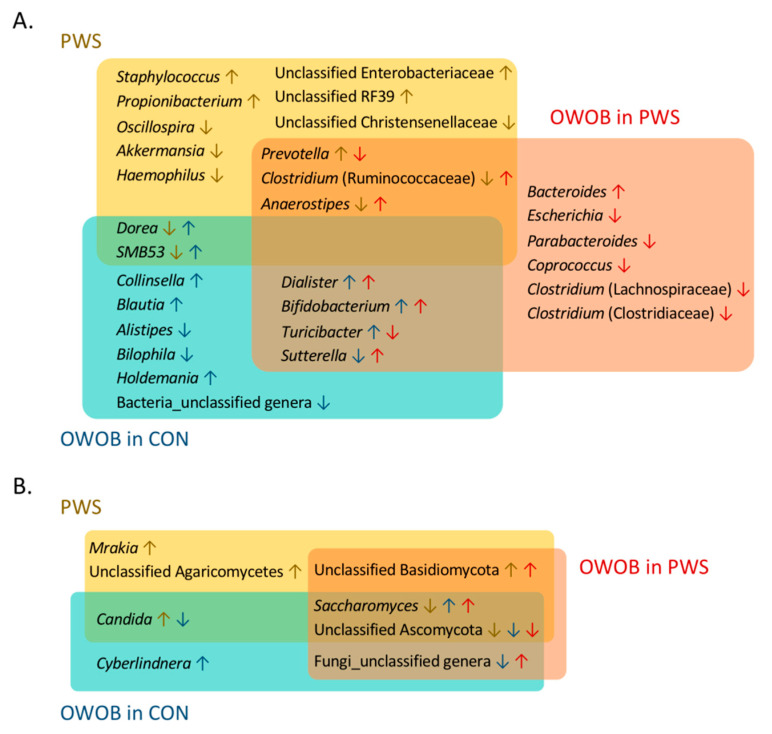
Overlap of differentially abundant bacterial (**A**) and fungal (**B**) taxa detected in Random Forest models for the PWS, OWOB PWS, and OWOB CON groups. PWS: Prader–Willi syndrome; CON: control; NW: normal weight; OWOB: overweight or obese. Upwards arrows indicate higher abundance, whereas downwards arrows indicate lower abundance than the reference group(s) (PWS vs. CON; OWOB in CON vs. NW in CON; OWOB in PWS vs. NW in CON).

**Figure 4 genes-11-00904-f004:**
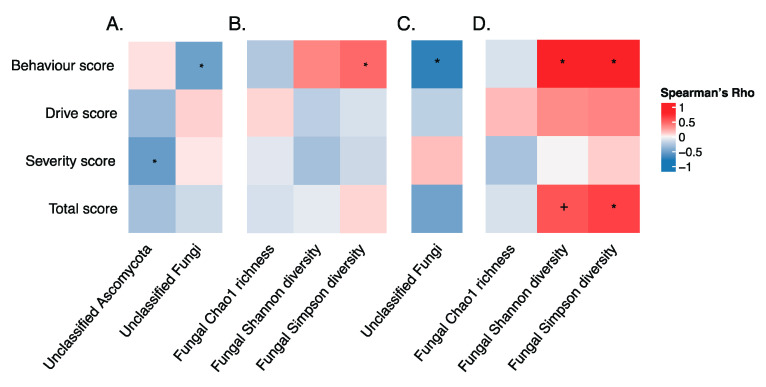
Correlation between hyperphagia and microbiota. (**A**) Correlation of hyperphagia score with fungal genus level in the whole dataset. (**B**) Correlation of hyperphagia scores with fungal species-level diversity in the whole dataset. (**C**) Correlation of hyperphagia score with fungal genus-level abundance within the PWS group. (**D**) Correlation of hyperphagia scores with fungal species-level diversity within the PWS group. The *q*-values were generated using the Bonferroni method: + indicates *q*-values between 0.05 and 0.1; * indicates *q*-values less than 0.05.

**Figure 5 genes-11-00904-f005:**
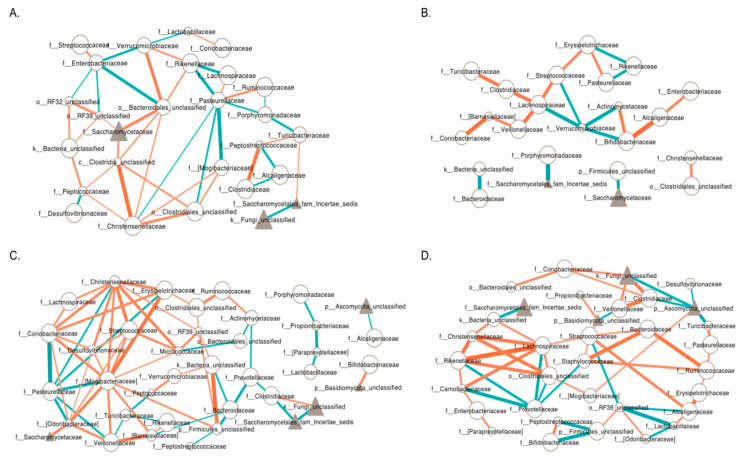
Microbial family (bacterial and fungal) co-occurrence networks for four subgroups. (**A**) Normal weight (NW) controls. (**B**) Overweight or obese (OWOB) controls. (**C**) NW PWS children. (**D**) OWOB PWS children. Taxon pairs with Spearman’s Rho ≥ 0.3 or ≤ −0.3 with *p*-values < 0.05 were visualized. Circular nodes represent bacterial families, and triangular nodes represent fungal families. The size of the nodes indicates the occurrence frequency in each subgroup. Green lines indicate negative correlations, and orange lines indicate positive correlations. Connector line thickness represents the value of the Spearman correlation coefficient (ρ). Family names on the nodes represent the proposed taxonomy by the Greengenes database.

**Table 1 genes-11-00904-t001:** Participant Characteristics.

	PWS (*n* = 25)	CON (*n* = 25)	*p*-Values
Sex (F/M)	14/11	9/16	0.256
Age (years)	6.2 (5.2, 12.9)	8.8 (6.3, 10.5)	0.455
BMI %ile	79.3 (65.5, 94.1)	76.6 (51.2, 91.5)	0.655
Weight status (OWOB/NW)	10/15	8/17	0.769
Hyperphagia scores **	19 (16, 26)	15 (14, 18)	0.014 *
Protein (g)	71 (65, 76)	64 (56, 73)	0.071
Carbohydrate (g)	189 (149, 206.3)	225 (193, 240)	0.002 *
Sugar (g)	198 (179, 212)	203 (184, 224)	0.441
Dietary fiber (g)	19 (16, 23)	17 (14, 21)	0.168
Fat (g)	204 (184, 211)	202 (196, 209)	0.848
SatFat (g)	195 (192, 203)	198 (195, 204)	0.147
UnSatFat (g)	376 (368, 390)	373 (362, 388)	0.386
Cholesterol (mg)	179 (107, 306)	163 (109, 267)	0.848

F: female; M: male; PWS: Prader–Willi syndrome; BMI %ile: body mass index-for-age percentile; OWOB: overweight or obese; NW: normal weight. Data presented as Median (25th and 75th percentiles). Comparison between the CON and PWS groups—continuous data: Wilcoxon test (two-sided); categorical data: Fisher’s exact test. * *p* < 0.05. ** Score ranging from 12 to 39 for the PWS group and from 12 to 25 for the control group (minimum possible score for the hyperphagia questionnaire is 11/55).

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
