# Peer review of "The Gut Microbiota Profile in Children with Prader–Willi Syndrome"

_genes, 2020, doi:10.3390/genes11080904_

Round 1
Reviewer 1 Report
While I would accept the paper as is, if any editing is required it would be appropriate to reiterate that this is a cross sectional study limiting any interpretation of the results, particularly with respect to hyperphagia.
Author Response
Thank you for your suggestion. We have added the suggested content to the manuscript on page 12: “We also note that the cross-sectional design limited interpretation of the results, particularly with respect to hyperphagia”.

Reviewer 2 Report
This is a very thorough exploratory study of the gut microbiota in a small sample of children and adolescents with PWS. The authors analyzed the microbiota diversity, composition and networks with respect to several parameters such as food intake (3d diary), BMI status, age, hyperphagia score and genotype. Although this is not the first report on the microbiota of individuals with PWS, this study expands the current knowledge in the field, and adds new analyses including an evaluation of the fungal community in these samples.
Overall, the manuscript is very well written, providing appropriate context and interpretation as well as acknowledging the limitations of the study design.
The authors evaluate their findings in comparison to that of Olsson et al, who examined the gut microbiota of obese individuals with PWS, but they should also consider their findings in the context of the recent publication by Garcia-Ribera and colleagues, who examined diet and microbiota composition in a cohort of children and adolescents with PWS in Spain (doi: 10.3390/nu12041063). This can be incorporated into the comparison to Olsson’s findings in the Discussion section.
Author Response
Thank you. We have added the suggested content to the manuscript on Page 10: Garcia-Ribera et al. also recently analyzed the fecal microbiota composition in children with PWS and observed higher phylogenetic diversity in normal-weight subjects compared to those overweight or obese [22]. Our subgroup analyses however found no differences in microbial diversity between OWOB PWS and NW PWS. Zhang et al. and Olsson et al. analyzed the microbial differences between PWS patients and healthy controls. In their studies, there was no comparison between subjects with overweight/obesity and those with normal weight. As such, when referring to Garcia-Ribera et al., we could only discuss the results of our study’s subgroup analyses (OWOB PWS vs. NW PWS).